# Egg Freshness Prediction Model Using Real-Time Cold Chain Storage Condition Based on Transfer Learning

**DOI:** 10.3390/foods11193082

**Published:** 2022-10-05

**Authors:** Tae Hyong Kim, Jong Hoon Kim, Ji Young Kim, Seung Eel Oh

**Affiliations:** 1Department of Biomechatronic Engineering, College of Biotechnology and Bioengineering, Sungkyunkwan University, Suwon 16419, Korea; 2Food Safety and Distribution Research Group, Korea Food Research Institute, Wanju-gun 55365, Korea

**Keywords:** egg freshness, haugh unit, real-time cold chain, weight loss, convolutional neural network, transfer learning

## Abstract

Maintaining and monitoring the quality of eggs is a major concern during cold chain storage and transportation due to the variation of external environments, such as temperature or humidity. In this study, we proposed a deep learning-based Haugh unit (HU) prediction model which is a universal parameter to determine egg freshness using a non-destructively measured weight loss by transfer learning technique. The temperature and weight loss of eggs from a laboratory and real-time cold chain environment conditions are collected from ten different types of room temperature conditions. The data augmentation technique is applied to increase the number of the collected dataset. The convolutional neural network (CNN) and long short-term memory (LSTM) algorithm are stacked to make one deep learning model with hyperparameter optimization to increase HU value prediction performance. In addition, the general machine learning algorithms are applied to compare HU prediction results with the CNN-LSTM model. The source and target model for stacked CNN-LSTM used temperature and weight loss data, respectively. Predicting HU using only weight loss data, the target transfer learning CNN-LSTM showed RMSE value decreased from 6.62 to 2.02 compared to a random forest regressor, respectively. In addition, the MAE of HU prediction results for the target model decreased when the data augmentation technique was applied from 3.16 to 1.39. It is believed that monitoring egg freshness by predicting HU in a real-time cold chain environment can be implemented in real-life by using non-destructive weight loss parameters along with deep learning.

## 1. Introduction

Eggs are one of the most representative perishable foods and are indispensable in human diets. They contain natural sources of essential nutrients, such as proteins and various vitamins [1]. Maintaining and monitoring the freshness of perishable food is a major concern in cold chain [2]. The variation of external environmental conditions (such as temperature, which is one of the most important factors in cold chain) is dynamic and unpredictable which will lead to various damages to perishable food [3]. However, during the transportation of eggs after laying, the egg aging process changes the physiological, chemical, nutritional and functional features of eggs, for instance, the loss of moisture evaporated through the shell pores and the escape of CO2 from albumen, which decreases the quality of egg during storage [4,5]. Therefore, it is important to monitor the condition of egg quality during the cold chain process to preserve the optimal condition of perishable food throughout the entire logistics process for the safety of the consumer [6]. 

The major parameter for defining egg quality is freshness. Egg quality or freshness is reflected by the change in storage time, temperature and humidity and decreases due to the deterioration of internal quality [7]. Egg freshness can be estimated by measuring several egg characteristics such as albumen height, Haugh unit (HU), albumen and yolk pH, etc. [8]. There are non-destructive and destructive methods to measure egg characteristics to estimate egg freshness. A HU is a standard destructive measurement to estimate the internal quality of an egg which is calculated by the albumen height and weight of the egg. One of the internal quality parameters of an egg, such as yolk pH, which can be measured by destructive methods, shows relatively high HU prediction performance since it is directly related to the HU value. However, this method needs to break eggs and can only be carried out in laboratory conditions, which is time-consuming and resource-wasting [9]. In comparison, non-destructive methods such as spectral analysis or near-infrared (NIR) technology have become popular due to their portability and ability to monitor freshness on-site. Lin et al. predicted HU using NIR spectroscopy with a correlation coefficient of 0.87 [10]. In addition, Cruz–Tirado et al. predicted and classified the egg freshness parameter, HU, using a portable NIR device with a relative error of 7.3% [11]. However, this non-destructive method of measuring egg quality using a hand-held NIR device showed low HU prediction performance. It has limitations that require an additional calibration process and must equip devices to achieve results [12]. In addition, expensive or non-portable devices have limitations for application in real-time cold chain transportation processes. Therefore, the development of a convenient way of estimating egg freshness value is essential.

Several researchers reported the methods to estimate HU value using various egg freshness parameters, such as laboratory condition temperature, humidity, albumen height, yolk pH or images based on partial least square regression and image processing technique [13]. Yimenu et al. developed the multi-variable kinetic regression model for egg freshness HU value prediction using temperature data [14]. Recently, a machine learning technique was applied to the food quality detection research field [15]. Nematinia et al. applied an artificial neural network (ANN) to predict egg freshness using various internal and external quality parameters, such as yolk height and pH, yolk density, HU and area index. The results showed a 0.93 correlation coefficient between measured and predicted HU values [16]. In addition, Cruz-Tirado et al. applied support vector machine (SVM) regression and a classification algorithm to predict HU value and classify the freshness of eggs using multiple parameters with a relative error of 7.32% and 87.0% accuracy [11]. 

Current egg freshness quality estimation methods using regression or machine learning algorithms are based on multiple numbers of egg freshness parameters measured using destructive methods in laboratory experimental conditions. These measured parameters used to predict HU value, especially temperature or humidity, shows high variation due to external environmental conditions, such as weather which leads to low HU prediction performance. For instance, the temperature changes when the cold-chain door is open and closed frequently or transporting eggs during the summer and winter seasons show a high variation of humidity. In addition, a few researchers have used multiple temperature conditions or real-time cold chain temperature conditions to predict HU values [17]. However, there are no previous studies to predict HU values using only weight loss data in real-time cold chain conditions.

Therefore, the aim of this study is to develop a deep learning-based HU prediction model using a single variable based on the static and dynamic temperature data. The deep learning model is constructed by a convolutional neural network (CNN) and long short-term memory (LSTM) algorithm. The method proposed in this study shows high prediction performance, even with weight loss parameters which can be gathered by non-destructive methods through transfer-learning.

## 2. Materials and Methods

### 2.1. Data Acquisition

Fresh-laid eggs were sampled and obtained directly from Ireefarm Egg Company in Seoul and the experiment is carried out at the Korea Food Research Institute. The egg samples were stored in a temperature-controlled storage refrigerator in the center of a storage room with 10 different types of temperature conditions. First, the storage room temperature was set at 1 °C, 5 °C, 10 °C, 15 °C, 20 °C, 25 °C and 30 °C constantly. Second, the storage room temperature changed from 10 to 20 °C and 10 to 30 °C every 24 hours dynamically. Last, the storage temperature changed under real-time temperature conditions in cold chain transportation [14]. The constant and dynamic changes in temperature are selected to mimic the egg storage at room temperature. The storage relative humidity condition was set as 50~90% [4]. This is accomplished by tracking actual shipments of eggs from farm warehouses transported to retail display at supermarkets. Egg sampling was performed to measure weight and albumen height to calculate weight loss and HU. During each sampling time, two egg cartons (10 eggs) were randomly selected for HU and weight loss. For the weight loss measurement process, eggs were labeled from 1 to 20 on the initial day of the storage experiment.

### 2.2. Apparatus

Current temperature monitoring systems used in the cold chain that can follow the product during storage and transport, such as strip chart recorders or temperature dataloggers, are usually expensive and not automated, thus requiring manual inspection [18]. Time–temperature indicator (TTI) or smart tag is a simple, inexpensive device that can attach to the package surface and integrate the cumulative time–temperature history of foodstuff, fully or partially. TTIs are effective in the realization of the cold chain throughout food processing and distribution. It provides a visual summary of a product‘s accumulated cold chain history, recording the effects of both time and temperature, to monitor, record, and cumulatively indicate the overall influence of temperature history on the food product quality from the point of production up to the consumer. For each temperature condition, the data logger (HL-1D/TL-1D, Rotronic Measurement Solutions, Bassersdorf, Swiss) is placed in the chamber to record the surface temperature every 10 minutes [8]. Next, the weight of (g) and albumen height (mm) using a digital scale and digimatic indicator, respectively, were measured from one egg carton from a group of cartons, selected randomly, for each temperature condition. Next, an HU is calculated for obtained egg information depending on storage temperature (Equation (1)).
(1)$$Haugh\;Unit=100\log\left(H+7.51-1.7*W^{0.37}\right),$$
where H is the albumen height and W is the weight of the egg.

Next, a weight loss is calculated based on the weight (in grams) measured from each labeled egg using an analytical balance. A digital scale (Model CSG 201F, OHAUS Corporation) was used (Figure 1). The labeled egg is placed on a plastic holder to measure weight. Egg weight loss is determined as the difference between the weights of the egg on different weighing days. The difference between day 0 and each measurement date was expressed as a percentage of the day 0 [14].

### 2.3. Data Processing

#### Data Sampling

Data preprocessing procedure is carried out. First, the temperature data carries a noise, and the moving average filter is applied to remove noise. Second, the size of temperature data for each sample has a different length, even if the data logger is set to the same storage duration. Therefore, there is a normalization process to match the data size for each temperature condition. As mentioned above, an HU is measured every 7 days for 1 °C; 5 days for 5, 10 °C; 3 days for 15, 20; 2 days for 25, 30 °C; 1 day for 10 to 20 and 10 to 30 °C, respectively (Figure 2 and Figure 3). In addition, an HU is measured every day for real-time cold chain conditions as shown in Figure 4 [7,8]. During the data sampling process, there are at least ten cartons in one group and one egg is destructed to measure the parameters for the calculation of HU. Therefore, we increased the dataset which was destructed by adding random values between −0.3 to +0.3 degrees, which is a standard deviation from a set of temperature data loggers.

As mentioned above, the calculated HU value is a discrete manner, as shown in Figure 5a. For instance, a temperature condition with 1 °C measured HU every 7 days, 15 times. Therefore, the number of HU and weight loss data are 15 (Figure 5b). To match the size of the temperature data, the cubic interpolation is applied for the HU and weight loss parameter, as shown in Figure 6.

### 2.4. Deep Learning-Based HU Prediction Model

In this research, we propose the HU value, which is the egg freshness parameter, using a transfer-learning technique based on the stacked CNN-LSTM deep learning model. The stacked CNN-LSTM model uses the non-destructive egg freshness measurement method, which is weight loss by transferring an important feature extracted from CNN from the source model of CNN-LSTM, which uses temperature data for training. The proposed transfer-learning-based HU prediction model flowchart is shown in Figure 7.

#### 2.4.1. Data Augmentation

Deep learning requires a large dataset for successful classification or predictable results. However, the acquisition of a dataset, such as the HU of an egg, is laborious work and it’s not always possible to have enough dataset. The performance of deep learning can be improved by applying the data augmentation technique [19]. There are a wide variety of other interesting methods that previous researchers applied, such as geometric transformation (e.g., rotation).

In this research, we applied four data augmentation techniques: (i) jittering, (ii) scaling, (iii) pooling and (iv) warping. First, we randomly divide the original dataset into two groups where one group is for data augmentation and the other group of data is used for testing the deep learning model. For the jittering technique, white gaussian noise with a measured value of 62 and 64 was applied. Next, a scaling technique was used with scaling factors of 0.9 and 1.1. Then, the pooling is applied with a pooling window size of 4 and 8, respectively. Lastly, the warping technique was applied with a warping factor of 0.9 and 1.1 to increase the data size, respectively.

Prior to the data augmentation, the original number of datasets was 2750, which is sampled from 2750 cartons. The number of static and dynamic room temperature condition datasets are 1850 and real-time cold chain temperature conditions are 900, respectively. In this study, the number of datasets are split into training and testing datasets with a split ratio of 50/50 (1375/1375). After applying the data augmentation technique, the number of datasets for training increased from 1375 to 6875 (Original dataset (1375) + white gaussian noise dataset (1375) + scaling dataset (1375) + pooling dataset (1375) + warping dataset (1375)). The final number of the augmented dataset, which is used as training for deep learning, is 6875 and the testing dataset is 1375, respectively. The data augmentation results are shown in Figure 8. The raw temperature signal is represented as a black line and the augmented signal is represented as red.

#### 2.4.2. Deep Learning

The aim of deep learning is to solve deep structure learning, one of the limitations of existing artificial neural networks, while simultaneously utilizing deep neural networks. It has been actively used in deep learning images, text, and voice recognition areas. Recent studies have reported that deep learning methods are also being applied to food quality detection fields [20,21]. The most important characteristic of the deep learning model is that it learns hierarchical features on its own. Learning hierarchical features here means that the first layer of the model is mainly learning to extract general features, whereas the closer the last layer of the model is, the more advanced learning is to extract specific features that can only appear in a particular dataset or problem [22].

#### 2.4.3. Convolutional Neural Network

CNN is a type of deep learning model which is designed to automatically learn spatial hierarchies of features, from low to high-level patterns [23]. A typical CNN consists of two parts. The first is the convolutional base. This is where multiple convolutional layers and pooling layers are stacked, and this part effectively extracts features from images or data. The convolutional layer calculates the dot product of two metrics. The first matrix is the learnable parameters and the second matrix is the kernel value. The result obtained from the calculation is an activation map and is calculated by the following equation (Equation (2)).
(2)$$G\left[m,n\right]=\left(f*h\right)\left[m,n\right]=\sum_{j}\sum_{k}h\left[j,k\right]f\left[m-j,n-k\right],$$
where f is the input matrix and h is the kernel function.

Next is the pooling layer. This is the blocking block of one CNN architecture, which serves to down-sample the input, reduce the dimensionality, and reduce computational effects. Next is the classifier part. It is mainly composed of fully connected layers, which means that neurons in all layers are connected to the output nodes of the previous layer. The goal of this part is to learn the extracted features well and classify them into categories [24]. There are several well-known pre-trained deep learning CNN architectures, such as AlexNet, ResNet, DenseNet, CifarNet, etc. In this research, the Residual Network-18 (ResNet-18) architecture is applied, and the detailed architecture of ResNet-18 is shown in Figure 9b. The key to ResNet-18 architecture is to use the Residual block. Unlike the previous pre-trained CNN architecture, this architecture creates a shortcut so that the input value can be added directly to the output value. ResNet basically has a VGG-19 backbone structure. This is a structure in which a shortcut is added after a convolution layer is added. ResNet-18 used a convolution filter sized 3-by-3 for the rest of the convolution layer except for the first convolution layer, and doubled the depth of the feature map when the size of the feature map was reduced by half. In addition, to reduce the size of the feature map, the size of the convolution slide was set to 2 instead of the pooling layer [25].

#### 2.4.4. Stacked Long Short-Term Memory

In this research, one of the deep learning algorithms, the bi-directional LSTM model, is stacked with the ResNet-18 CNN model to predict HU using a temperature and weight loss parameter. LSTM is a modified version of a recurrent neural network which has the advantage of predicting sequential time-series data or specific values. The backpropagation through time and gradient descent is used to train the model [10,25]. The LSTM unit is constructed by a memory cell and three main gates (input, output and forget). Based on this structure, it can manage which information should be forgotten and remembered to learn long-term dependencies. In order to stack multiple LSTM layers, several LSTM layers are stacked by the memory state, and the hidden state of each layer is forwarded as input to the next layer [26,27]. The architecture of the initial stacked bidirectional-LSTM model is composed of a sequence input layer with a size of 1 (temperature or weight loss), 2 bi-directional LSTM layers with a hidden unit of 250, a dropout layer with a value of 0.2, a fully connected layer with 32 neurons and an output regression layer with one neuron.

To stack the bi-directional LSTM model followed by CNN, the additional layers, such as sequence unfolding and folding layers, need to be placed as shown in Figure 9a.

#### 2.4.5. Transfer Learning

Recently, deep learning models have been actively developed, but the process to train them from “scratch” is time-consuming. It requires securing a lot of data, and computational capabilities. As mentioned above, the method developed to overcome this problem is transfer learning. The concept of transfer learning is to use relatively complex and well-trained models from large data sources. Wang et al. reported ImageNet which uses large data sources to build pre-training models and transfers the learned knowledge for classification to other models [28]. In addition, transfer learning has been widely applied to various fields of application, such as medical imaging, manufacturing, and human activity recognition. With this technology, new models are being developed through fewer datasets, shorter training times, and improved performance [29].

In order to use transfer learning, it is necessary to select which layer is frozen or updated when transferring from the previously learned source model to the target model. There are general steps and rules to applying transfer learning. After pretraining the source model, go through the fine-tuning procedure to apply it to the target model. This is to obtain good initial weights through pre-training of the source model [30]. In this study, we applied a fine-tuning transfer learning method that “replaces” the last fully connected layer, classification, or regression layer, as a new classification or regression layer, respectively, as a method of utilizing features extracted from a pre-trained network.

As mentioned above, there are source and target models to predict HU value using a deep learning model via transfer learning. The source model was proposed to predict the HU value based on a temperature condition dataset and the target model was composed of a weight loss dataset. In Figure 9b, the ResNet-18 is composed of four main residual blocks. In this study, the specific layer to be transferred to the target model is selected based on fine-tuning methods. The rest of the layer, which is selected to not transfer (freeze), is defined by applying a layer learning rate of 0. The training option for the target model is also transferred from the source model.

### 2.5. Hyperparameter Optimization

Methods for determining parameters include manual searching and optimization. In general, manual searching methods performed through trial-error methods require professional knowledge and skills, and there is room for user randomness to intervene, making it difficult to secure objectivity and reliability. On the other hand, since the optimization method estimates the parameters of the outflow model relatively efficiently and objectively using numerical algorithms, various studies using the optimization method are being conducted [31,32].

Hyperparameters of all models used in this study were optimized using the Bayesian optimization method. The Bayesian optimization method is used as an efficient method to find the optimal solution for a function with a large amount of computation based on the Gaussian probability [33]. As an optimization objective function of the proposed architecture, it is constructed based on the root mean square error (RMSE) of the regression model, which is shown in Equation (3). In addition, the CNN-LSTM combination model used RMSE as the objective function.
(3)f(x)=min(RMSE),
where *f(x)* is the optimization objective function and error is the value calculated based on the RMSE of the regression model.

The initial training options for the stacked CNN-LSTM model are listed as follows: (1) training function: ADAM, (2) gradient threshold: 1 (3) learning rate: 0.001, (4) l2regularization rate: 0.001, (5) batch size: 64. The maximum optimization iteration was set to 30. Table 1 represents the training hyperparameter variables and their ranges optimized by the Bayesian optimization method.

### 2.6. Machine Learning

In order to compare and validate the performance of the proposed stacked CNN-LSTM-based deep learning model via transfer learning, two general machine learning techniques from previous studies have been used [34]. The regression models used are as follows: ANN regressor and random forest (RF) regressor. For ANN regressor, the input data is composed of a weight loss parameter and output data is HU value from real-time cold chain data. The initial structure of the ANN regressor is composed of two hidden layers with a hidden node size of 50, respectively. The training function is set as ‘trainlm’. The parameters for training ANN are learning rate, number of hidden node sizes, and training function. For the RF regressor, the input and output data are identical to the ANN regressor. The parameters for training RF regressor are the number of trees, the number of leaf size and the number of predictors to sample. The optimal parameters for ANN and RF are searched by using the Bayesian optimization technique with a maximum iteration of 35. The objective function of optimization is to minimize the normalized RMSE which is identical to the stacked CNN-LSTM deep learning model optimization objective function.

In this study, temperature and weight loss data were used in training/testing ratios of 50/50 for a machine learning model, which is identical to a deep learning model, respectively. The performance was evaluated using data augmented training dataset 5 repetition. The general machine learning and transfer-learning-based deep learning model were constructed using MATLAB R2021b version (The Mathworks, Inc., Natick, MA, USA) and RTX 3080 Ti GPU.

### 2.7. Data Analysis and Performance Measure

The performance of the proposed transfer-learning based HU prediction model and two general machine learning algorithms are evaluated. The prediction model of HU value using non-destructive method parameters, which are temperature and weight loss value, was evaluated with RMSE, maximum absolute error (MAE), and Pearson correlation coefficient (R) using a deep learning stacked CNN-LSTM model. The equations for RMSE, MAE, and R are shown in (4)–(6), respectively.
(4)$$RMSE=\sqrt{\frac{1}{N}\overset{N}{\underset{i=1}{\sum }}{\left(y_{i}-y_{i}^{*}\right)}^{2}}\;\;\;\;\;\;$$
(5)$$R=\frac{\sum_{i=1}^{N}\left(y_{i}-\bar{y_{i}}\right)\left(y_{i}^{*}-\bar{y_{i}^{*}}\right)}{\sqrt{\sum_{i=1}^{N}{\left(y_{i}-\bar{y_{i}}\right)}^{2}{\left(y^{*}{}_{i}-\bar{y_{i}^{*}}\right)}^{2}}}$$
(6)$$MAE\left(\%\right)=\frac{\;HU{\;}_{Model}-HU{\;}_{ref}}{HU{\;}_{ref}}$$
where y_i_ represents the actual value, y_i_* represents the predicted value from the model, and N represents the length of 1 trial. In Equation (2), R represents the value of the Pearson correlation coefficient. The actual value is y_i_ and y_i_ is the value predicted by the model, max(y_i_) is the maximum value and min(y_i_) is the minimum value. The significance level was set as *p* < 0.01.

## 3. Results

The HU of an egg during static, dynamic, and real-time cold chain conditions is predicted using a stacked CNN-LSTM model, ANN and RF regressor using temperature or weight loss data, respectively. As shown in Figure 10, the predicted vs actual HU value using temperature data by RF and ANN regressor showed an R-value of 0.76 and 0.75 for no augmentation, respectively. This study applied a data augmentation technique while training the machine learning model. The RMSE value of ANN and RF regressor with a temperature dataset when data augmentation is applied showed 13.8% and 8.1% lower than no augmentation model.

In Figure 11, the general machine learning regressor model using weight loss data to predict HU value showed an average r value of 0.56, 0.68, 0.62, and 0.76 for ANN and RF with no data augmentation and data augmentation, respectively. In addition, the MAE value for the weight loss data model using ANN and RF were 6.97, 5.76, 5.13 and 4.99 with no data augmentation and data augmentation, respectively. The results showed higher RMSE values of 17.5% and 8.7% as compared to temperature data for ANN and RF with data augmentation technique, respectively. In general, the performance of predicting HU value was higher for the temperature data model as compared to the weight loss data model for both ANN and RF, respectively.

As shown in Figure 12, the transfer learning-based stacked CNN-LSTM regressor showed a high R-value for the source model (temperature data only) and the target model (weight loss data only) of 0.95 and 0.86, respectively. In addition, the r value of the source and target model were 0.98 and 0.96 when data augmentation is applied, respectively. The source model showed RMSE and MAE values of 2.04 and 1.27, and 1.45 and 0.92 when data augmentation is not applied and is applied, respectively. The RMSE value of the target model decreased by 53% when data augmentation is applied (Table 2).

As mentioned above, the target model of the stacked CNN-LSTM model transferred weight from the source model, and the target model showed higher RMSE and MAE from 1.27 to 2.02 and 0.92 to 1.40 with the data augmentation technique as compared to the source model. However, the performance (RMSE, MAE and R-value) was outperformed even when the weight loss data was used for training and testing as compared to ANN and RF with weight and temperature data (Table 2). As compared to the ANN machine learning models, the RMSE and MAE source models of the transfer learning model decreased by 272.05% and 259.31 %, and the R-value was increased by 21.05%. In addition, for the target model of the stacked CNN-LSTM model, the RMSE and MAE decreased from 9.19 to 4.39 and 9.67 to 3.17 and the R-value increased from 0.56 to 0.87 where no data augmentation is applied. Overall, the target model of the stacked CNN-LSTM model showed a lower R-value of 0.96 whereas ANN with weight loss and RF with weight loss model showed an R-value of 0.62 and 0.76 when data augmentation is applied, respectively (Table 3).

## 4. Discussion

Freshness is a major contribution to the quality of an egg and it’s necessary to determine the freshness of an egg systematically to achieve uniform quality since consumers can misunderstand changes in freshness as a lack of quality. Therefore, it is very important to monitor the freshness of the egg during the storage and transportation process. In this study, the deep learning-based HU prediction model is proposed to predict HU using weight loss data in a non-destructive way.

There are several indices to evaluate egg freshness, such as yolk index, albumen pH, air chamber diameter and height, etc. Previous research reported that the HU is the index commonly used to evaluate egg freshness [13,16]. Most previous studies predicted that the HU value used temperature data since the freshness or quality of food is highly influenced by the storage time and conditions [5,35,36]. Akter et al. reported that the storage condition fluctuates by a variation of external factors, mainly temperature, humidity and time, which affect the HU of the egg [36]. In this study, we demonstrated that the weight loss has the potential to predict HU value instead of temperature. The weight, which is one of the parameters to calculate HU, can be measured without breaking the egg and can be applied to real-time conditions. However, according to the author’s knowledge, there are no previous studies that used only weight loss values to predict HU. Brasil et al. reported that the prediction of HU using partial least squares regression and support vector machine regression showed r^2^ values of 0.71 and 0.79, respectively which is lower than our HU prediction results from using a portable NIR spectrometer [1]. Therefore, the transfer learning is applied to increase performance, even with weight loss data as shown in Table 2 where the transferred target model RMSE and MAE value decreased from 6.09 to 2.02 and 4.59 to 1.39 as compared to the ANN regressor. In future, the parameters such as humidity and seasonal temperature can be applied to our source model to better predict performance [37].

In this study, we proposed an HU prediction model. The CNN is stacked with LSTM models, along with a transfer learning technique. There are several machine learning models, such as autoregressive integrated moving average (ARIMA) or adaptive neuro-fuzzy inference system (ANFIS) to predict values such as food temperature. These models show a relatively high prediction performance when the input data is stationary, and when there is less fluctuation or a shorter period of time data. However, the performance of these algorithms is questionable when they are implemented in real-time cold chain environments due to the high variance and high level of noise in raw signals. Many previous studies implemented deep learning models such as CNN or LSTM, or stacked CNN with LSTM algorithms to increase performance [38,39]. The CNN algorithm can automatically extract effective features from the raw signal data instead of a manual feature extracting process, and LSTM can find the interdependence of time series data and automatically detect the best mode suitable for relevant data [37]. In addition, the combined CNN-LSTM model showed a higher R-value and a lower RMSE and MAE while predicting HU as compared to ANN and RF regressor (Table 2). This result cannot be directly comparable to other previous research; however, it shows a similar pattern in which the performance is generally better for both temperature and weight data-based models [40]. The transfer-learning base combined the CNN-LSTM model and can be applied to various food freshness quality monitoring systems, such as milk or beef during storage where a simple or portable device is required in real-life.

## 5. Conclusions

Overall, we proposed a novel method to predict the HU value which is the major parameter to evaluate egg freshness using a non-destructive weight loss parameter. We constructed a deep learning model with a transfer learning technique to increase prediction performance. In future, this model can allow monitoring of the freshness of eggs in real-time cold chain and consumers can determine how fresh the eggs are at the counter.

## Figures and Tables

**Figure 1 foods-11-03082-f001:**
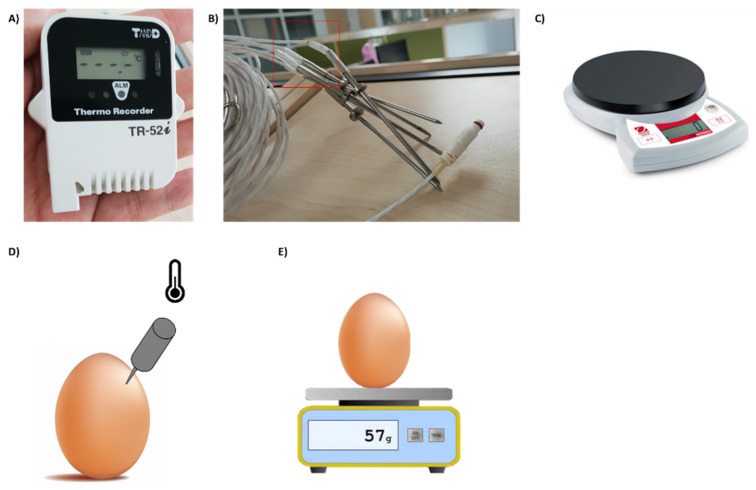
The apparatus for measuring temperature and weight of egg: (**a**) the temperature measurement device; (**b**) the temperature measurement tip; (**c**) the digital scale; (**d**) temperature measurement method; (**e**) egg weight loss measurement method.

**Figure 2 foods-11-03082-f002:**
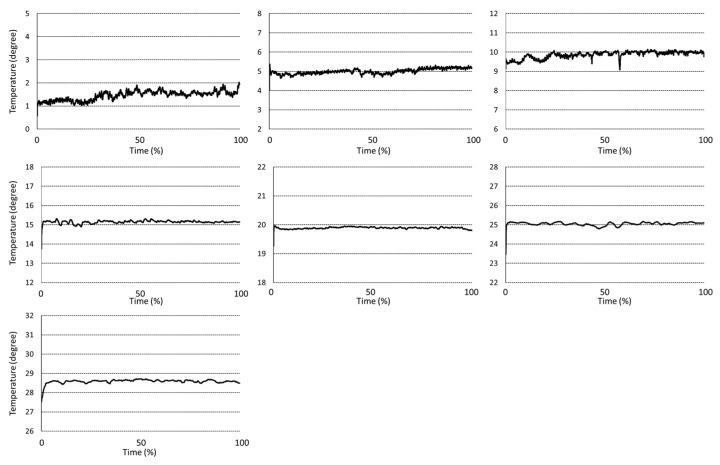
Static temperature condition for measurement of egg freshness during storage.

**Figure 3 foods-11-03082-f003:**
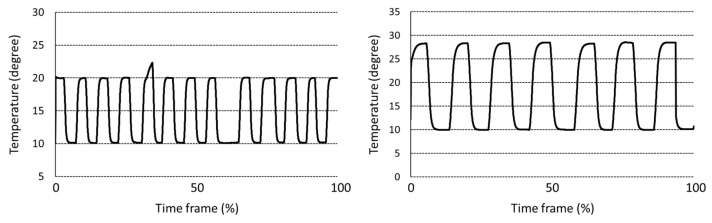
Dynamic temperature condition for measurement of egg freshness during storage.

**Figure 4 foods-11-03082-f004:**
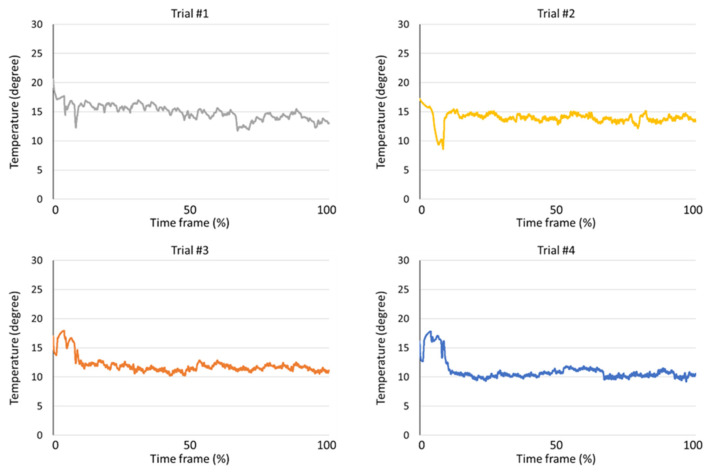
Real-time cold chain temperature condition for measurement of egg freshness.

**Figure 5 foods-11-03082-f005:**
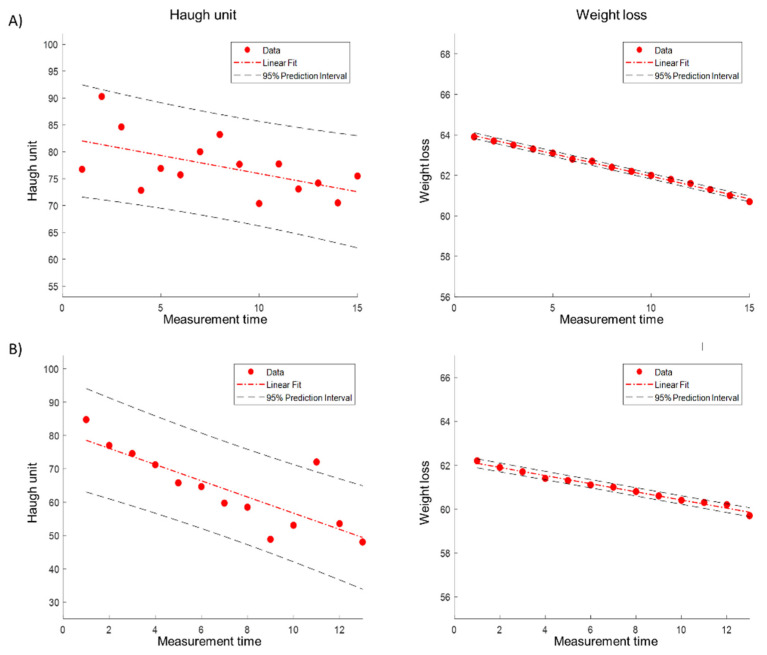
Calculated HU and weight loss of egg for dynamic and real-time cold chain temperature conditions: (**a**) the haugh unit and weight loss during real-time cold-chain; (**b**) the haugh unit and weight loss during static temperature condition.

**Figure 6 foods-11-03082-f006:**
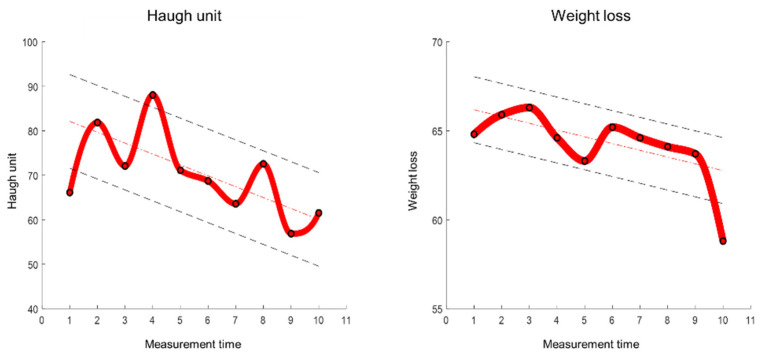
Interpolation of HU and weight loss of egg during real-time cold chain temperature condition.

**Figure 7 foods-11-03082-f007:**
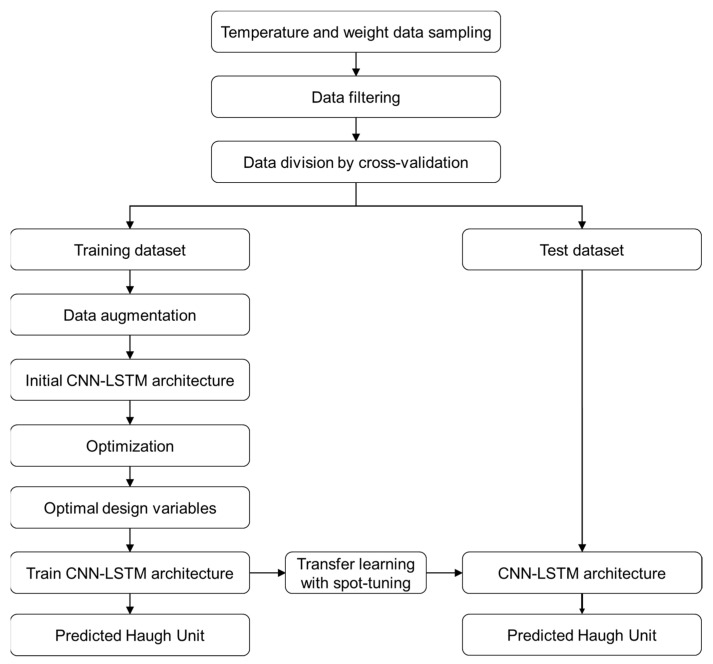
The flowchart of the transfer-learning-based HU prediction model.

**Figure 8 foods-11-03082-f008:**
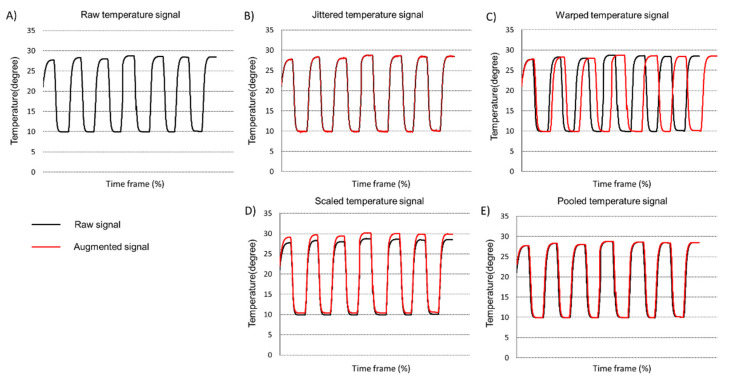
Implementation of four different data augmentation techniques to raw temperature signal: (**a**) raw temperature signal; (**b**) jittered data augmented signal; (**c**) time-warped data augmented signal; (**d**) amplitude scaled data augmented signal; (**e**) pooled data augmented signal.

**Figure 9 foods-11-03082-f009:**
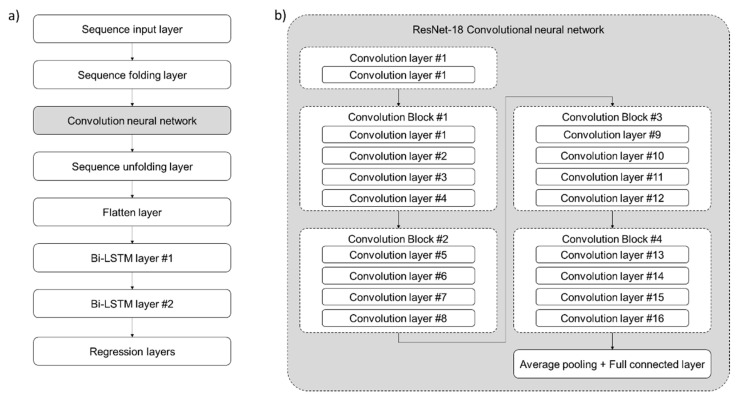
(**a**) The architecture of stacked CNN-LSTM deep learning model; (**b**) ResNet-18 CNN architecture.

**Figure 10 foods-11-03082-f010:**
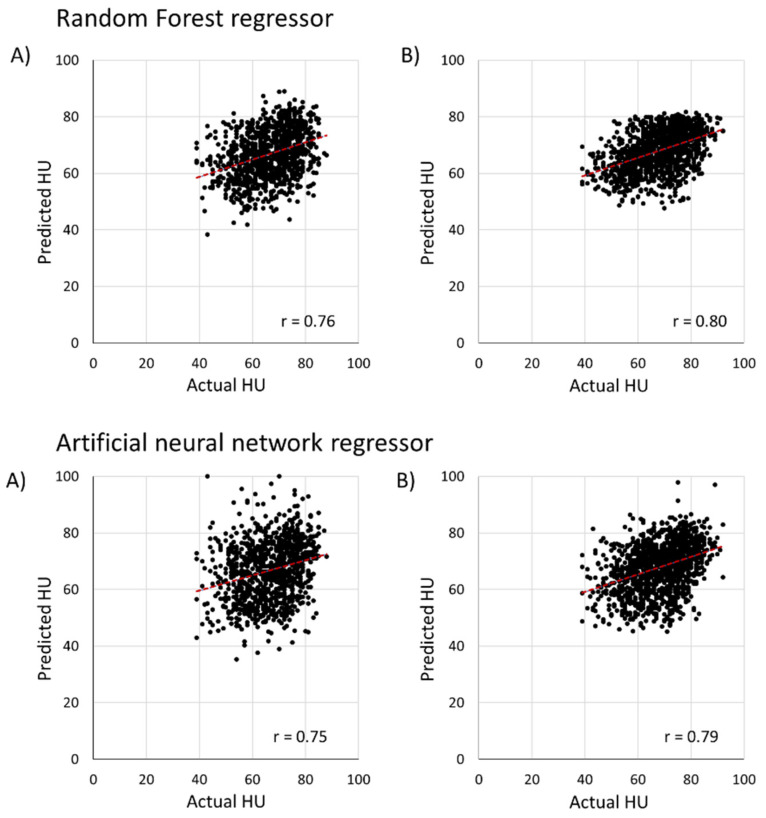
The result of general machine learning (RF and ANN regressor) for predicting HU using temperature data: (**A**) the results without data augmentation for random forest and artificial neural network regressor; (**B**) represents the results with data augmentation for random forest and artificial neural network regressor.

**Figure 11 foods-11-03082-f011:**
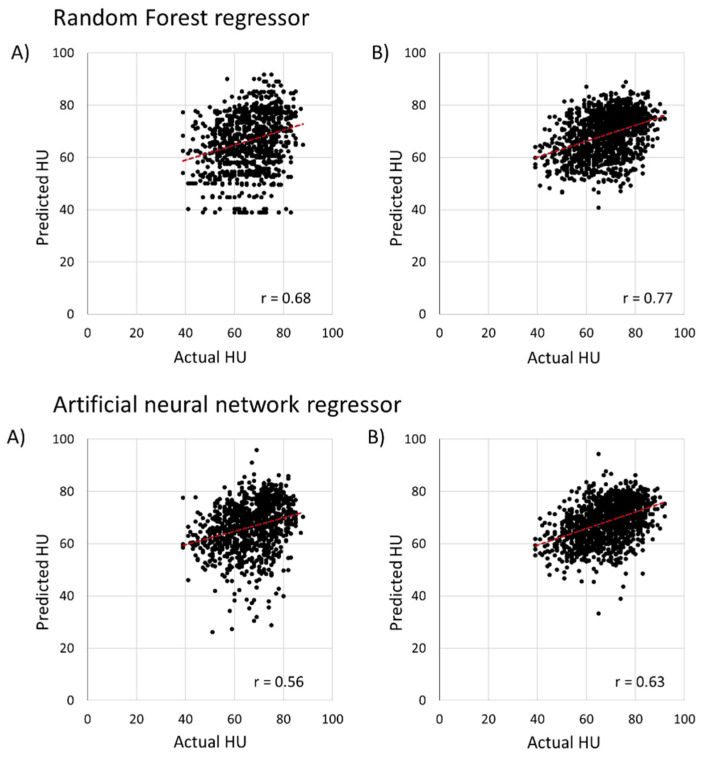
The result of general machine learning (RF and ANN regressor) for predicting HU weight loss data: (**A**) the results without data augmentation for random forest and artificial neural network regressor; (**B**) represents the results with data augmentation for random forest and artificial neural network regressor.

**Figure 12 foods-11-03082-f012:**
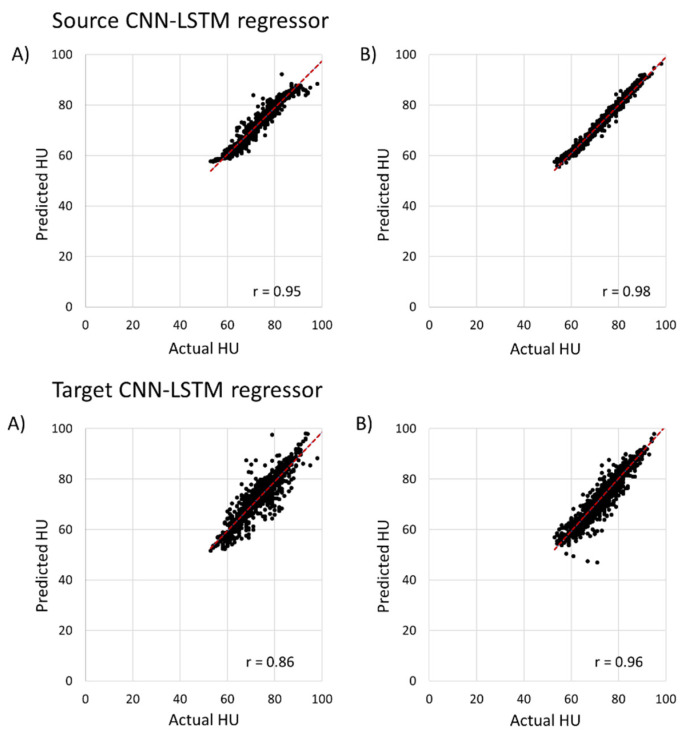
The result of a transfer-learning-based CNN-LSTM regressor model for predicting HU using temperature and weight loss data. The source CNN-LSTM regressor is only used for temperature data and the target CNN-LSTM regressor used only weight loss data: (**A**) the results without data augmentation; (**B**) represents the results with data augmentation.

**Table 1 foods-11-03082-t001:** The list of hyperparameter options for training combined CNN-LSTM deep learning model.

Parameter	Values
Epochs	[25, 50]
Dropout rate	[0.1, 0.2, 0.3, 0.4, 0.5]
Mini batch size	[8, 16, 32, 64]
Gradient threshold	[0, 1, 2, 3, 4, 5, 6]
Gradient threshold method	[Global-l2norm, l2norm]
Learning rate	[0.00001, 0.0001, 0.001, 0.01]
Optimizer	[ADAM, SGDM, RMSProp]
Moment	[0.6, 0.7, 0.8, 0.9]

**Table 2 foods-11-03082-t002:** The HU prediction result of transfer-learning based stacked convolutional neural network with long short-term memory regressor using temperature and weight loss data.

Performance	Model Type	No Data Augmentation	Data Augmentation
RMSE	Temperature with ANN	7.59 ± 0.29	6.67 ± 0.15
	Temperature with RF	6.55 ± 0.30	6.06 ± 0.17
	Weight loss with ANN	9.19 ± 0.56	8.08 ± 0.34
	Weight loss with RF	7.44 ± 0.33	6.64 ± 0.21
	Temperature CNN-LSTM	2.04 ± 0.29	1.27 ± 0.37
	Weight loss CNN-LSTM	4.39 ± 1.14	2.02 ± 0.35
MAE	Temperature with ANN	5.21 ± 0.27	4.96 ± 0.11
	Temperature with RF	4.22 ± 1.78	4.59 ± 0.10
	Weight loss with ANN	6.97 ± 0.24	5.13 ± 2.26
	Weight loss with RF	5.76 ± 0.31	4.99 ± 0.17
	Temperature CNN-LSTM	1.45 ± 0.21	0.92 ± 0.27
	Weight loss CNN-LSTM	3.17 ± 0.74	1.40 ± 0.24
R	Temperature with ANN	0.75 ± 0.03	0.79 ± 0.01
	Temperature with RF	0.76 ± 0.02	0.80 ± 0.02
	Weight loss with ANN	0.56 ± 0.03	0.62 ± 0.02
	Weight loss with RF	0.68 ± 0.02	0.76 ± 0.02
	Temperature CNN-LSTM	0.95 ± 0.01	0.98 ± 0.01
	Weight loss CNN-LSTM	0.87 ± 0.06	0.96 ± 0.01

**Table 3 foods-11-03082-t003:** The overall result of general machine learning and transfer-learning-based CNN-LSTM model using temperature and weight loss data with and without data augmentation technique.

Model Type	Data Augmentation	Performance	Trial #1	Trial #2	Trial #3	Trial #4	Trial #5	Average
Temperature with ANN	No DA	RMSE	8.13	7.26	7.55	7.47	7.56	7.59
		MAE	5.73	5.11	5.03	5.01	5.19	5.21
		R	0.70	0.78	0.76	0.75	0.76	0.75
	DA	RMSE	6.79	6.83	6.71	6.62	6.41	6.67
		MAE	5.13	4.99	4.96	4.97	4.76	4.96
		R	0.77	0.78	0.80	0.79	0.81	0.79
Temperature with RF	No DA	RMSE	6.74	6.01	6.76	6.47	6.79	6.55
		MAE	5.23	0.67	5.11	4.88	5.2	4.22
		R	0.74	0.79	0.74	0.77	0.74	0.75
	DA	RMSE	5.91	5.99	5.90	6.2	6.32	6.06
		MAE	4.54	4.47	4.53	4.68	4.75	4.59
		R	0.81	0.80	0.82	0.77	0.79	0.79
Weight with ANN	No DA	RMSE	9.03	9.11	10.01	9.48	8.32	9.19
		MAE	7.01	6.96	7.09	7.27	6.53	6.97
		R	0.57	0.54	0.54	0.57	0.61	0.56
	DA	RMSE	8.14	7.73	7.79	8.68	8.09	8.08
		MAE	6.44	6.09	6.08	6.29	6.21	5.13
		R	0.62	0.65	0.64	0.59	0.63	0.62
Weight with RF	No DA	RMSE	7.31	7.46	7.09	8.07	7.31	7.44
		MAE	5.70	5.81	5.33	6.29	5.7	5.76
		R	0.69	0.67	0.72	0.65	0.69	0.68
	DA	RMSE	6.82	6.77	6.84	6.42	6.36	6.64
		MAE	5.08	5.07	5.22	4.78	4.82	4.99
		R	0.75	0.75	0.74	0.79	0.78	0.76
Source CNN-LSTM	No DA	RMSE	2.10	2.26	1.69	1.71	2.44	2.04
		MAE	1.49	1.65	1.22	1.22	1.71	1.45
		R	0.93	0.96	0.97	0.97	0.95	0.95
	DA	RMSE	1.03	1.83	1.59	1.07	0.84	1.27
		MAE	0.79	1.34	1.12	0.78	0.58	0.92
		R	0.98	0.97	0.97	0.98	0.99	0.98
Target CNN-LSTM	No DA	RMSE	3.74	4.42	4.93	6.15	2.74	4.39
		MAE	2.65	3.49	3.30	4.30	2.11	3.17
		R	0.90	0.88	0.81	0.79	0.95	0.87
	DA	RMSE	2.35	1.73	1.65	1.86	2.53	2.02
		MAE	1.61	1.17	1.13	1.36	1.73	1.40
		R	0.95	0.97	0.97	0.97	0.94	0.96

## Data Availability

The data presented in this study are available on request from the corresponding author.

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
