# Peer review of "Egg Freshness Prediction Model Using Real-Time Cold Chain Storage Condition Based on Transfer Learning"

_foods, 2022, doi:10.3390/foods11193082_

Round 1

Reviewer 1 Report

The manuscript is written with clear understanding of the project addressed. However, there are some concerns that need to be addressed to enhance the quality of the manuscript. My specific comments are as follows:

Abstract:

Elaborate more on the methods of study.

Introduction:

“Previous research reported that the loss of water and carbon 42 dioxide through the egg shell and the osmotic exchange…” Explain those research

“First, a destructive method which has high accuracy however, this method needs to break eggs and..” what are the examples of destructive method?

“Several researchers reported the methods to estimate HU value with various egg freshness parameters such as..” Explain those methods

Based on your objectives, please compare how your study is different from those that have already been published

Methods:

“First, the storage temperature was set at 5℃, 10℃, 20℃ and 30℃ constantly. Second, the storage temperature changed from 10 to 20℃ and 10 to 30℃ every 24 hours dynamically. Last, the storage temperature changed under real-time temperature conditions in the cold chain.” So, how many storage temperature overall? Very confusing

“As mentioned above, a HU is measured every 7, 3, 2, 1, 2, 1 and 2 days for temperature of 1, 5, 10, 20, 30, 10 to 20 and 10 to 30℃ and..” why are the days are not consistent?is there any time/day interval for each storage temperature?

Deep learning section should comes first before data augmentation

What is split ratio for model training used for deep learning

Results and discussion:

Revise caption for figure 12

Transform figure 13 to table, so we could see the trend better

“Previous research reported that the HU is the internationally-used evaluation index.” What research? Justify

“Most of previous research who predict HU value used temperature data since the freshness or quality of food is highly influenced by the storage time and conditions.” add citation

“There are several algorithms to predict a specific value such as autoregressive integrated moving average (ARIMA) or adaptive neuro-fuzzy inference system (ANFIS).” What value?

The findings lack in terms of justification

Conclusions:

Add conclusion section

Add recommendation for future studies

General comments:

Please check the reference styles and grammar of the manuscript.

Reviewer 2 Report

Maintaining and monitoring egg quality during storage with influence of temperature and humidity is an important research purpose. The manuscript investigated temperature and weight loss during storage and utilized deep learning method to link it with environmental parameters, and reached a solid conclusion that the model can be used to predict egg quality. The study was with clear structure and the conclusion is supported by the results. English needs to be significantly improved. Some comments need to be addressed :

1. Revise the sentence in line 31-33 since there are two which

2. Line 62, please define low prediction ability, does that mean low prediction accuracy or complicated predicting process?

3. The introduction is needs to be condensed a bit

4. Would you like to explain the sampling process why 20 eggs were enough for validating the prediction results

5. Where were the 20 eggs located, how did you manage a random selection guaranteeing the locations of corners, centers of the room? How is the temperature and realtive humidity distribution in the room everywhere?

6. Can you add the manufacturer of the smart tag or TTI used in this study in line 113?

7. Line 347, should r be R value?

8. Line 347: before and after applying data augmentation

9. The conclusion is too long which needs to be more concise, focusing only on the key findings of the current study without repeating the background information and results.

Reviewer 3 Report

1-I could not understand these expressions. What did you mean? Line 403-404

‘Egg is the one of main protein source which can easily found in our life. Due to consuming less fresh egg, the consumer can be expose to food poison especially in summer season.’

An expression like this would be better.

‘Eggs are one of the few foods that are used throughout the world; thus the egg industry is an inqiortant segment of the world food industry. Eggs have been an important part of the human diet since the dawn of recorded history. In modem times eggs have been an important commodity in international trade.’

https://www.routledge.com/Egg-Science-and-Technology/Stadelman-Newkirk-Newby/p/book/9781560228554

2- The statements given in the discussion section could have been clearer.

‘Freshness makes a major contribution to the quality of egg and egg products. One of the most important objective in the food industry is that of achieving a uniform quality both of raw materials and of the final product. One of the main concerns of the egg industry is the systematic determination of egg freshness, because consumers may perceive variability in freshness as lack of quality.’

https://www.researchgate.net/publication/226567838_Methods_to_evaluate_egg_freshness_in_research_and_industry_A_review

3-Since the subject is eggs, I would enter with eggs. I used to talk about eggs being a delicate food. Line 29-30

4-What are the quality characteristics of eggs? What are the freshness determination methods? Who did what on these issues? What are the new techniques? What exactly is the method you use? The answers to these questions are very scattered. A more regular flow would be good. Line 28 to 93

5- While conveying the result, what is done is given as can be done. Line 342-343

‘This section may be divided by subheadings. It should provide a concise and precise description of the experimental results, their interpretation, as well as the experimental conclusions that can be drawn.’
